# Visual analytics framework for survival analysis and biomarker discovery from gene expression data

Jaka Kokošar[1], Cagatay Turkay[2], Luka Ausec[3], Miha Štajdohar[3], Blaž Zupan[1,4]*

1 Faculty of Computer and Information Science, University of Ljubljana, Ljubljana, Slovenia, 2 Centre for Interdisciplinary Methodologies, University of Warwick, Coventry, United Kingdom, 3 Genialis Inc., Boston, Massachusetts, United States of America, 4 Department of Education, Innovation and Technology, Baylor College of Medicine, Houston, Texas, United States of America

* blaz.zupan@fri.uni-lj.si

## Abstract

We introduce a visual analytics methodology for survival analysis, and propose a framework that defines a reusable set of visualization and modeling components to support exploratory and hypothesis-driven biomarker discovery. Survival analysis—essential in biomedicine—evaluates patients' survival rates and the onset of medically relevant events, given their clinical and genetic profiles and genetic predispositions. Existing approaches often require programming expertise or rely on inflexible analysis pipelines, limiting their usability among biomedical researchers. The lack of advanced, user-friendly tools hinders problem solving, limits accessibility for biomedical researchers, and restricts interactive data exploration. Our methodology emphasizes functionality-driven design and modularity, akin to combining LEGO bricks to build tailored visual workflows. We (1) define a minimal set of reusable visualization and modeling components that support common survival analysis tasks, (2) implement interactive visualizations for discovering survival cohorts and their characteristic features, and (3) demonstrate integration within an existing visual analytics platform. We implemented the methodology as an open-source add-on to Orange Data Mining and validated it through use cases ranging from Kaplan–Meier estimation to biomarker discovery. While the framework is generally applicable, we illustrate its value through case studies in cancer research, where survival analysis is of critical importance. The resulting framework illustrates how methodological design can drive intuitive, transparent, and effective survival analysis.

## 1 Introduction

Survival analysis is a set of statistical methods used to determine the life expectancy of an investigated population. Survival analysis became particularly valuable in medical research, where researchers use survival data to observe the life expectancy

**Data availability statement:** The toolbox we developed is available as an open-source add-on "Survival" to Orange (http://orangedatamining.com). The code is available at https://github.com/biolab/orange3-survival-analysis. The workflows from the case studies, together with the data, are available at https://orangedatamining.com/examples under the section "Survival Analysis".

**Funding:** This work was supported in part by grants from Slovenian Research Agency (P2-0209, L2-3170, V2-2272).

**Competing interests:** The authors have declared that no competing interests exist.

of the investigated population. Notably, in a clinical setting, we refer to survival time as a time to some clinical outcomes, like relapse or progression of a disease [1,2]. In biomedicine, these covariates are understood as potential biomarkers and targets for treatment development (see Fig 1). To improve survival outcomes, it is essential to understand how patients with different health conditions and genomic predispositions respond to various treatments [3]. Furthermore, the identification of novel markers is crucial for the development of personalized medicine, a rising trend in evidence-based medicine [4].

Survival analysis tools should therefore support the inference of survival, the study of changes in survival over time, and the identification of cohorts of subjects with substantial differences in survival. Cohorts are characterised by features, and tools should support users in finding and exploring the most informative ones. Most of these functionalities are currently available through libraries in scripting environments such as R and Python (e.g., `survival` [5], `scikit-survival` [6], and `lifelines` [7]), where they integrate well with other data visualisation and modelling techniques. However, due to the lack of computer programming experience, these are difficult to access for biomedical domain experts. Moreover, scripting does not lend itself to exploratory data analysis and does not facilitate communication with the domain experts involved.

These limitations of scripting environments for survival analysis call for a complementary visual analytics approach, which combines interactive visualizations, integrated analysis methods, and visual programming. Visual analytics [8] enhances data exploration by allowing users to interactively manipulate and interpret data through graphical interfaces, while interactive visualizations [9] provide dynamic, real-time feedback to reveal patterns and trends. Visual programming (e.g., [10]) enables intuitive, workflow-based analysis, reducing the need for coding and making complex methods more accessible—an essential advantage for medical data analysis, where interpretability, usability, and collaboration between clinical researchers and data scientists are critical.

Here, we report on our visual analytics approach to survival analysis. We aim to design an intuitive but powerful tool with a smooth learning curve and interactive interface that motivates exploratory data analysis, explanation, and discovery, supporting a broad scope of potential applications with as few components as possible. The main contributions of our work thus include:

1. **Task-oriented decomposition of survival analysis.** We define and implement a reusable set of analytical components that together cover key survival analysis goals. This decomposition supports flexible construction of diverse analytical pipelines and facilitates exploratory biomarker discovery.

2. **Demonstration of concept validity through replication and application.** We show that with the designed components, users can replicate recently published biomedical survival analyses and address complex, large-scale datasets—demonstrating both the generality and practical utility of the framework.

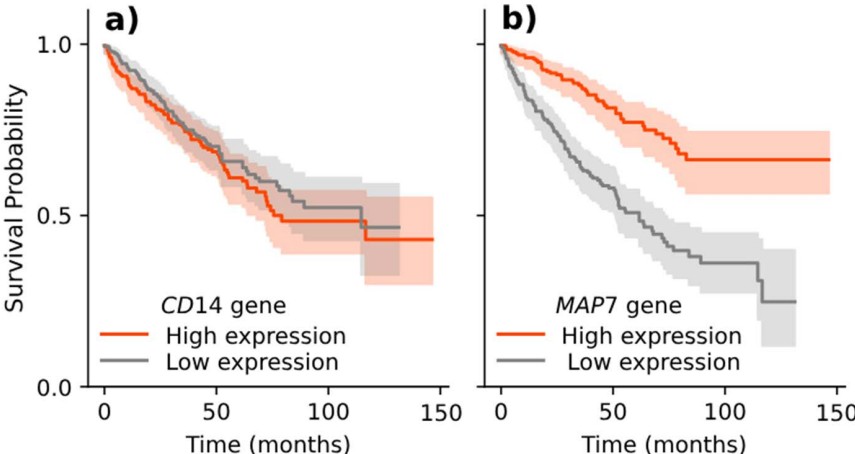

**Fig 1. An example of a Kaplan-Meier survival plot for two gene expression-dependent conditions associated with patient survival. (a)** The difference between survival curves of gene *CD14* is not evident. **(b)** The survival curve is substantially higher for a group of patients with highly expressed gene *MAP7*. We could say that *MAP7* is hence a better biomarker for survival. In biomarker discovery, one of the tasks is to rank features, e.g., gene mutation status, gene expression values, and protein levels, according to the degree of separation between survival signatures given gene expression.

3. **Integration with a broader data mining and machine learning ecosystem.** Implemented as an add-on to Orange Data Mining (see https://orangedatamining.com) [11,12], the framework integrates seamlessly with a wide range of existing visual programming tools, enabling the complementary use of advanced data visualization, machine learning models, and interaction analysis beyond survival-specific methods.

Our visual analytics framework and all workflows presented in this paper are freely available as open-source resources to promote transparency, reproducibility, and further community-driven development.

The manuscript begins with an overview of related methods, including survival analysis, visual analytics, and workflow-based approaches to data analysis. We then describe our visual analytics approach, which integrates interpreted workflows [13], interactive visualisations and the use of computational models. To demonstrate the usefulness of the designed analytical components and tool, we present a set of typical survival analysis and biomarker discovery tasks and show how we can solve them using our proposed visual analytics toolbox.

## 2 Related work

In application, the data analysis framework we present here relates to concepts from survival analysis and discovery of biomarkers. In the development of exploratory data analysis approach, our work is related to visual analytics and development of pipelines that use workflows and visual programming, and combine machine learning and interactive data visualization. In particular, for the latter, our work borrows from concepts developed within Orange Data Mining framework.

### 2.1 Survival analysis for biomarker discovery

Biomarkers play a pivotal role in linking a patient's molecular characteristics to clinical outcomes, helping to predict the success of therapies and guide decision-making. In biomedicine, the covariates with a significant impact on patient survival are potential markers, characteristics of a biological system we can measure objectively and use as an indicator of the system's state [14]. In oncology, for instance, biomarkers can distinguish patients likely to benefit from specific therapies, while sparing others from ineffective treatments. Markers may be clinical, related to the patient's symptoms, or biological, related to measurements on the molecular level, like the concentration of a specific protein or expression of a

particular gene [15,16]. Identifying such markers is crucial for the development of personalized medicine that uses data to deliver targeted, patient-specific treatments [17].

## 2.2 Key visual analytics concepts

Here we discuss a number of key visual analytics principles to facilitate data analysis tasks involving large numbers of data features and multiple data sources. We then highlight key relevant techniques and applications in related domains.

*Interplay of computation and human input* – Visual analytics is often referred to as a dialogue, an interplay, between the analysts and the data, through the facilitation of computational tools and interactive visualizations [8]. Interaction and integrated computational methods that are triggered by human inputs have been a key facilitator of dialogues that are mechanisms for analysis and reasoning [18]. For instance, such integrated approaches have been used to cluster large collections interactively [19], or for high-dimensional data analysis [20]. Through flexible programmable visual analytics workflows, our approach in this paper facilitates such interactions in a more structured manner and makes opportunities for the analysts to interact more explicit.

*Input variation & result comparison* – A key mechanism often used in visual analytics methods is to explore the variation in the results in response to variation of data and parameters that control an analytical operation or a model [21]. Visual analytics approaches have shown to effectively support comparative analysis for modelling in the context of classification [22], or in dimension reduction [23]. In this work, we build on this notion of input-output variation exploration within the context of survival analysis.

*Visual analysis workflow & interactive propagation* – A number of visualization approaches and systems have made the analysis workflow explicit and available for further interaction starting with the seminal VisTrails work by Bavoli et al. [24]. More recently, similar visual programming and visual workflow methods have been used in generating visualization design variations [25]. While these methods are primarily focused on visualization generation, they pave the way for more general, flexible and transferable data analysis workflows where interactive alterations at an intermediate stage of the analytical process can be propagated further through the analytical tools and visualizations that communicate the results.

## 2.3 Visual analytics for survival analysis

The challenges around survival analysis have been approached by visual analytics methods in a number of studies. In their Caleydo system, Lex et al. demonstrates how Kaplan-Meier (KM) plots could be associated as visual indicators to characterize sub-cohorts in a visual analysis of cancer subtype stratification process [26]. Corvo et al. makes survival analysis a core analysis focus and demonstrates how KM plots could be generated in response to interactive selections in their SurviVIS tool [27]. Marai et al. employ KM plots and Nomograms in their visual analytics system designed to explore and detect similar head and neck cancer patients [28]. However, these solutions primarily focus on a pre-defined tasks (or disease types) and do not support user-defined analytical pipelines. Our work takes these methods further by integrating a combination of analytical methods and interactive visualizations within a visual programming paradigm.

Several web-based tools with a focus on survival analysis have been proposed to assist biomedical researchers in genome-based biomarker discovery [29,30]. These tools provide a graphical user interface combining genomics data, clinical data and analysis tools designed specifically for mining survival-related gene markers across different data sources and tumor types. For instance, Maller et al. used *Kaplan-Meier Plotter* to evaluate the association between *PLOD2* gene expression values and relapse-free survival [31] and Ingebriktsen et al. used it to evaluate a six-gene signature concerning recurrence-free survival in breast cancer patients [32]. While the proposed web-based solutions can make exploring data simpler, they do not fully implement the key concepts of visual analytics (discussed in Sect 2.2) and this dashboard-based approach often struggle to adapt to real-time insights or changing hypotheses. In addition, as discussed by Idogawa et al. [33], there are concerns about the data integrity of the proposed systems and more importantly, the

underlying analysis is often closed-source. This ultimately leads to a lack of trust within the research community and hinders the reproducibility of research results.

### 2.4 Orange data mining framework

We have implemented the visual analytics approach to survival analysis we describe here as an add-on to the Orange data mining framework [10,12,34]. Orange is a general-purpose data analysis tool. Data analysis pipelines in Orange are assembled through visual programming and consist of widgets—components with a graphical user interface—that perform distinct data analysis or visualization tasks. We entirely borrow the visual analytics concept from Orange. The principal innovation of the work presented here is designing components for survival analysis so that they can collectively be applied to a wide range of tasks and seamlessly integrate with other existing Orange components. Notice that we could develop a similar survival analysis framework in different general purpose workflow-based data analysis systems, including popular RapidMiner (see https://rapidminer.com/) or KNIME (see http://knime.com). We choose Orange, though, since it uses interpreted workflows [13], that execute in real time as users interact with individual components. In an interpreted workflow, every change to widget settings, data selections, or parameters is immediately propagated through the pipeline without requiring explicit re-execution. This property enables tightly coupled human–machine interaction and supports the exploratory, iterative nature of visual analytics. Orange is also different from the two mentioned workflow systems in supporting interactive visualizations. Despite all these differences, our choice for Orange is biased: several authors of this report are its principal designers and contributors.

## 3 Key analytical approaches to survival analysis for biomarker discovery

The literature shows that current practice uses a limited range of computational and data visualization methods for the analysis of survival data. These methods can be used in combination to address many different task in biomarker discovery where the overreaching goal is to identify or validate data features that are best associated with patient survival. By far the most widely used and established methods of survival analysis are the Kaplan-Meier estimator and Cox regression model. Their adaptation and importance are evident in their status among the most cited articles [35] as well as more focused reviews and guidelines on their practical use in biomedical cancer research [1,36–40].

### 3.1 Characterization of key goals and tasks

Based on the above-mentioned reviews and guidelines, we have externalized the common goals (**G**) and tasks (**T**) typically employed in different analytical scenarios. While we do not delve into the details and definitions of survival analysis methods, we do, however, outline their use in the context of biomedical application and how they can be applied across diverse analytical scenarios in oncology.

G1 – **Understand the distribution of survival times.** Survival data is most often presented as a survival curve, which illustrates how survival prognosis changes over time. The most fundamental step in survival analysis is the construction of the survival curve.

T1 **Estimate and plot the survival curve.** The most common approach to survival curve estimation is the Kaplan-Meier method [41]. The calculated survival probabilities are then plotted against time to provide a visual summary of survival probabilities over time.

T2 **Summarize key statistics.** Kaplan-Meier curves can be thought of as a form of descriptive statistics for survival data [42]. To add context to the visual representation, Kaplan-Meier curves are often juxtaposed by additional statistics, such as total number of samples, number of samples with observed events, median survival time, and confidence intervals for survival probabilities, among others.

G2 – **Analyze the differences in survival between different cohorts.** Survival curves provide a basis for comparing the survival experience of different sample groups.

 

T3 **Stratify the data by variables of interest.** Kaplan-Meier analysis does not directly account for the impact of various variables on survival. However, it is often used to conduct a univariate analysis for comparing survival curves of groups defined by different categorical variables. For continuous variables researches have to first define boundaries defining different groups.

T4 **Compare survival curves of different groups.** To inspect differences in survival between groups, researchers typically plot multiple survival curves on the same graph [42]. A visual inspection can provide an initial indication of whether a particular group has, for example, a more favorable prognosis compared to others. Visual comparison is further supported by statistical tests, predominantly log-rank test, to quantify this difference [43].

G3 – **Interpret the effect of data features on survival outcomes.** Survival analysis can be extended to measure the impact of different data features on survival rates.

T5 **Develop a prognostic model of patient survival.** Researchers often model the impact of data features on survival in either univariate or multivariate settings using Cox regression analysis [44]:

T5.1 **Univariate cox regression analysis** is often used a feature screening technique or to observe how each data features affects survival independently.

T5.2 **Multivariate cox regression analysis** is used to observe how the significance of features shifts when they are adjusted for other data features.

T6 **Model interpretation and explanation.** By examining the hazard ratios and confidence intervals derived from the prognostic models, the researchers assess the strength and direction of associations between data features and survival outcomes, identifying which factors have significant prognostic value.

T7 **Model application and visualization.** Researchers often present prognostic models graphically as Nomograms to enable individualized survival predictions [45]. Nomograms translate prognostic models into an accessible visual format that aids in their interpretation and makes it easier for clinicians to apply them in practice.

G4 – **Develop personalized risk-based patient stratification.** Patient stratification exemplifies the practical application of prognostic models, allowing identification of high-risk groups that may, for example, benefit from targeted interventions.

T8 **Assign risk scores to individual patients.** A common approach is to utilize the coefficients derived from the developed prognostic model to calculate the risk score (sometimes referred to as the prognostic index) for each patient [46]. These risk scores are then used to stratify patients into different risk groups. This stratification process is crucial for identifying subgroups of high- and low-risk patients with distinct prognoses that can inform personalized treatments or biomarker development and validation.

G5 – **Validation of developed prognostic or risk-based stratification models.** In biomarker discovery, models with strong discriminatory power can help validate the clinical relevance of the identified biomarkers, showing that they provide meaningful insight into patient outcomes [47].

T9 **Evaluating the discriminative power.** It is common practice to visually inspect the separation between the curves for high- and low-risk groups, which indicates good discriminatory power. More analytical another common approach is to use the concordance index (C-index) [48], which quantifies how well the model distinguishes between individuals who experience the event of interest and those who do not.

## 3.2 Typical analytical pipelines

This goal-oriented abstraction can generally be used for analyzing any type of survival data. However, typical biomarker discovery pipelines tend to be more complex and multifaceted. Also, how survival analysis is used can vary; survival data can be the primary concern of the analysis or just as a validation step, where the results are evaluated in relation to clinical outcomes. In either case, this goal-oriented abstraction can be applied across various analytical scenarios; each goal can function as a standalone analysis or be combined in different ways to suit specific analysis needs.

For example, in their clinical trail, Lawrence et al. [49] set out to evaluate the relationship between expression of the DNA mismatch repair (MMR) gene *MLH1* and survival outcomes in patients with resected pancreatic cancer who received adjuvant chemoradiation. They set out to analyze survival distributions of study population, evaluate the differences in survival of different cohorts, and assess the prognostic impact of potential gene marker (**G1**, **G2**, **G3**). Their data analysis pipeline includes stratifying patients based on median expression of *MLH1* (**T3**), estimating survival curves for each cohort (**T1,2**), evaluating how significant is the differance in survival between cohorts (**T4**) and estimating the prognostic value of gene *MLH1* in univariate and multivariate setting (**T5,6**). We can observe similar data analysis pipelines in several different studies for biomarker discovery in oncology [50–53].

Expanding on this concepts, in their analysis pipeline, Long et al. [54] developed and validated a multi gene-based prognostic index to stratify patients into low- and high-risk groups (**G4**, **G4**, **G5**). They first screened survival related genes (**T5.1**) and selected a four-gene prognosis model (**T5.2**) that assigns risk scores to individual patients. Their analysis includes a validation of newly developed prognosis model in relation to established clinical factors (**T6**) and construction of a predictive nomogram (**T7**). They also validate a risk-based patient stratification by splitting patients into low- and high-risk groups (**T3**), calculating survival curve for each group (**T1**) and test if there are significant differences (**T4**) in survival rates (**G1**, **G2**). Similary as before, we can observe this kind of analytical approaches across various different studies [55–57].

## 4 Integrating survival analysis, workflows and visual programming

Our approach to support survival analysis consists of decomposing the key analytical approaches (**T**) discussed in Sect 3 into independent and reusable components, each addressing a specific analytical or exploratory function. These components can then be assembled into diverse workflows through visual programming, allowing users to construct pipelines that meet the needs of their specific analytical scenarios (**G**). By grounding our design in this task-oriented approach, we have created a survival analysis framework that users can use to build custom data analysis pipelines ranging from simple exploratory analysis of survival curves to complex pipelines for multi-gene biomarker discovery, all within a visual programming environment. Below, we describe this general framework and list the components we have implemented for survival analysis.

### 4.1 Data analysis framework

The following elements from Orange were critical in supporting our design:

**Widgets** are Orange's components in data analytics workflows that receive data on input, process or visualize the data, and transmit the results or user selections to the widget's output. Widgets encapsulate data processing and visualization functions and allow us to combine tasks (**T**) and goals (**G**) into independent, reusable components:

**Kaplan-Meier plotter (T1-4)** is a component that we can use to analyse survival function (**G2**) and assess the differences of survival between different cohorts (**G2**).

**Cox Regression (T5-6)** component allows us to estimate the effect different data features have on survival of our patients (**G3**).

**Rank Survival Features (T5.1)** is a very specialized component that ranks data features based on univariate cox regression analysis.

**Cohorts (T8)** is what we can use for the development of prognosis model and assign risk score to each patient (**G4**).

**Survival Nomogram (T7)** enables graphical representation of developed prognostic model and supports its interpretation (**G3**).

We then combine these components into workflows using visual programming.

**Workflows** are ordered collections of widgets that combine different analytical tasks and goals into user defined analytical pipelines, like the ones discussed in Sect 3.2. For example, a workflow from Fig 2 starts with the widget

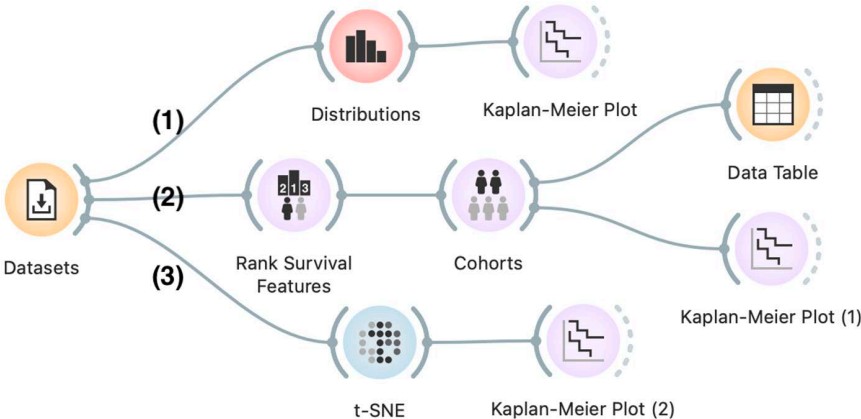

**Fig 2. An example workflow to explore cohort formation based on clustering and biomarker discovery.** The workflow starts by loading the data (*Datasets* widget). The workflow progresses from **(1)** stratification of patients based on specified cutoffs value of continuous features (using *Distributions*, *Kaplan-Meier Plot*) to more complex **(2)** cohort formation based on several data features (the branch with *Rank Survival Features*, *Cohorts*). For comparison, **(3)** we plotted the map of patients (*t-SNE*), where patients are grouped by similarity of their feature-profiles. The survival curve for the cluster of patients selected in t-SNE is then compared to all other patients (*Kaplan-Meier Plot (1)*).

for loading the data and then branches into pipelines for (1) exploring biomarker significance, (2) risk-based patient stratification, and (3) exploring similarity between data instances. This exemplifies how a visual programming paradigm supports iterative assembly of workflows, allowing users to test and refine their hypothesis in real time. Orange workflows are read from left to right, with inputs to widgets on the left and outputs on the right. Note that in this workflow *Kaplan-Meier Plot* widget is reused to display the difference in survival of either the feature-defined cohorts, cohorts defined by risk-based stratification model or between group of patients selected in the t-SNE visualization.

**Interactive interface.** Orange widgets are interactive. The first and most obvious type of interaction is the selection of display options in the widget's control area, as in the *Kaplan-Meier plot* widget from Fig 3a. More important for visual analytics, however, are interactive displays of data. For example, in the *Distributions* widget, the user can select any part of the distribution (Fig 3b) and instruct the widget to output only the subset of the selected data instances or pass selection indicator with the data on the output (Fig 3c).

**Signal processing.** To support interactive visual analysis, widgets emit their output whenever their data selections or parameters change. They communicate user's selections to downstream pipeline, where components instantaneously react to changes immediately. In Fig 2, for example in the middle branch, a combination of *Rank Survival Features*, *Cohorts* and *Kaplan-Meier Plot* provide a interactive way for users to select top-ranked data features that are then used for construction of risk-based model, split patients into low- and high-risk cohorts and plot the survival curve of each cohort (see Fig 4).

## 5 Evaluation through case studies

To evaluate our approach, we designed a series of case studies based on recently published research in survival analysis and biomarker discovery. These case studies test the ability of our approach and developed software to address key challenges in the field, assessing whether the provided components are sufficient, whether they can be effectively combined to solve relevant analytical tasks, and whether the approach is practical for real-world applications. Through these examples, we demonstrate how interactive visual analytics facilitates exploratory data analysis, hypothesis generation, and discovery of survival-related biomarkers.

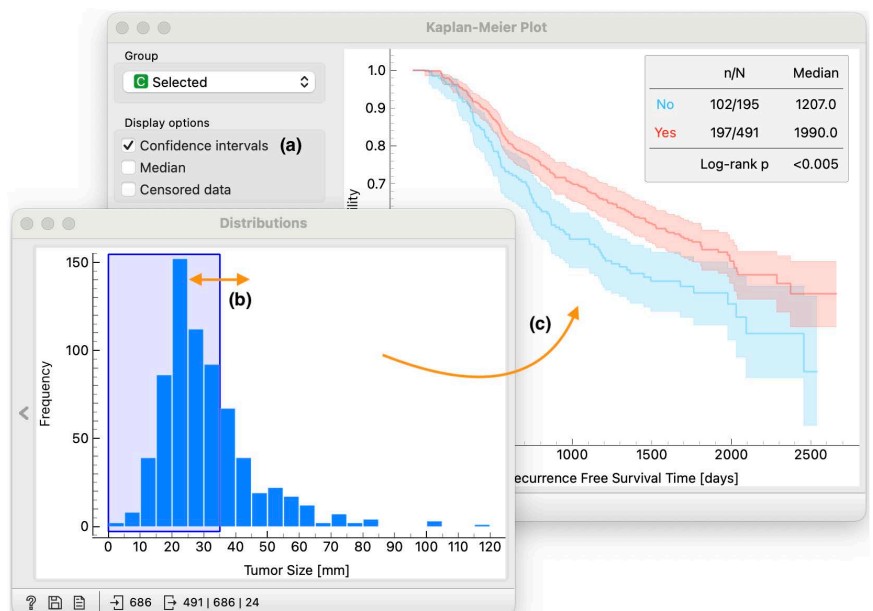

**Fig 3. We zoom in on the top branch in workflow from Fig 2.** We use the *Distribution* widget to plot the distribution of the tumor size feature. The distribution plot is interactive, for example, we select data instances where the tumor size is less than 35 mm **(b)**. Patient subgroups defined in the Distribution widget (b) are linked to survival curves in the *Kaplan–Meier Plot* **(c)** through brush-and-link interaction, providing an interactive browser to characterize the survival of the selected group. Different analysis options are available to the user, for example plotting of the confidence interval **(a)**.

## 5.1 Data

In all case studies, we use data from the METABRIC database. The METABRIC (Molecular Taxonomy of Breast Cancer International Consortium) dataset comprises genomic and clinical data from nearly 2,000 breast cancer patients, including expression profiles with approximately 24,000 genes, clinical features, and survival outcomes [58]. The dataset is publicly available and widely used in studies of breast cancer prognosis and biomarker discovery. We use this dataset as a representative example of a large, complex dataset with multiple data types and survival outcomes, which is typical of many studies in oncology and survival-related biomarker discovery. An example of the METABRIC data as provided in Orange is shown in Fig 5.

## 5.2 Exploring prognostic role of gene KRAS

Hwang et al. [51] have recently investigated the prognostic role of the mRNA expression of a proto-oncogene *KRAS* in breast cancer with the application of survival analysis on two different datasets (TCGA [59] and METABRIC [58]). The authors set out to find if the expression of *KRAS* is prognostic in breast cancer. *KRAS* is a well-known driver of lung, prostate, and colon cancer. To investigate the prognostic impact of *KRAS*, Hwang et al. stratified patients into high- and low-expression groups based on median expression levels (**T3**). They plotted Kaplan-Meier survival curves (**T1**) for overall survival of both cohorts and assessed statistical significance using log-rank test (**T4**). They incorporated multivariate Cox proportional hazards models to assess the independent prognostic value of *KRAS* while adjusting for key clinical variables, such as age, tumor stage, and receptor statuses (**T5-6**). The analysis showed significantly worse outcomes for the high-expression group and confirmed high *KRAS* expression as an independent indicator of poorer prognosis across both datasets.

In the workflow in Fig 6 we show how can we readily reproduce steps of their analysis with our proposed survival analysis components. Workflow starts with the *Genes* widget, a part of Orange's Bioinformatics add-on, we selected a

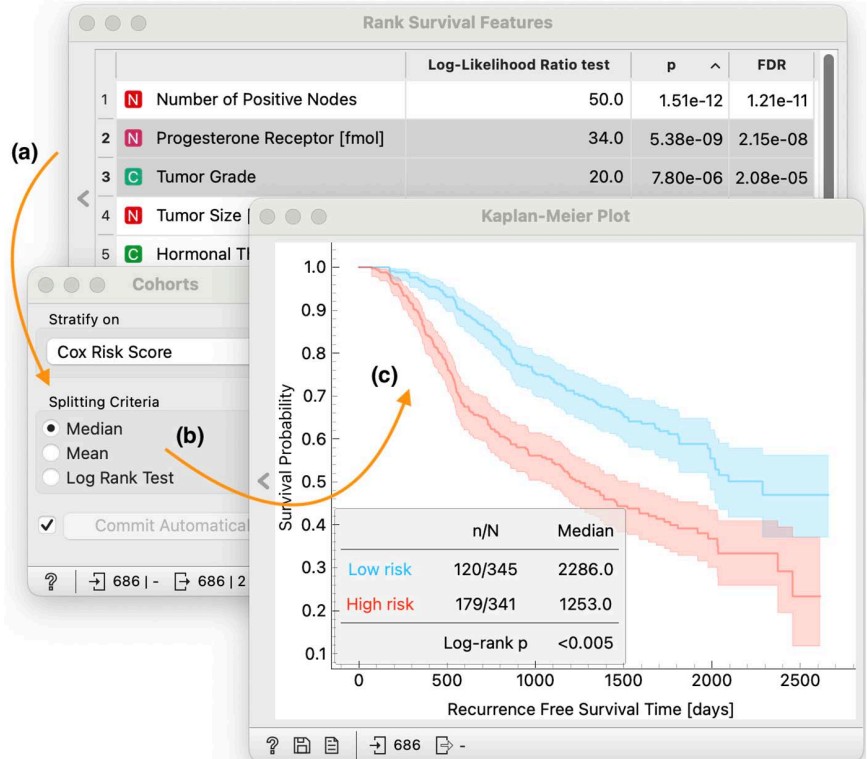

**Fig 4. Focusing on the middle branch of workflow from Fig 2.** We use the *Rank Survival Features* to prioritize features most relevant **(a)**. Data that is passed to *Cohorts* widget contains only two selected features to construct a risk-based model and stratify patients into low- and high-risk cohorts **(b)**. This information is then passed to the *Kaplan-Meier Plot* that estimates, plots and assess how significantly different are the two survival curves **(c)**.

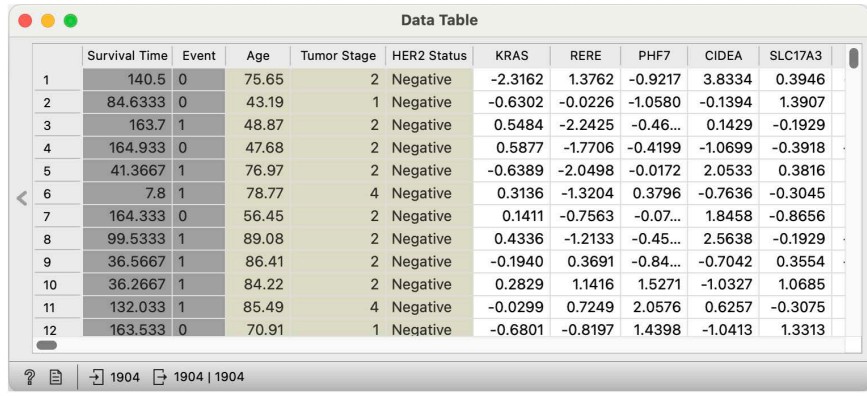

**Fig 5. A small sample of the METABRIC dataset available in Orange, where each row represents a primary breast cancer patient characterized by (1) clinical features (such as age at diagnosis and tumor stage), (2) gene expression values (e.g., *KRAS*, *RERE*, *PHF7*), (3) and clinical outcomes (overall patient survival).**

*KRAS* gene. The *Genes* widget also shows information from the Entrez Database—thus simplifying gene exploration within Orange. We use *Discretize* widget to specify the median expression as the threshold between high- and low-expressors. In the upper branch of the workflow we use *Kaplan-Meier Plot* widget to visualize and compare survival

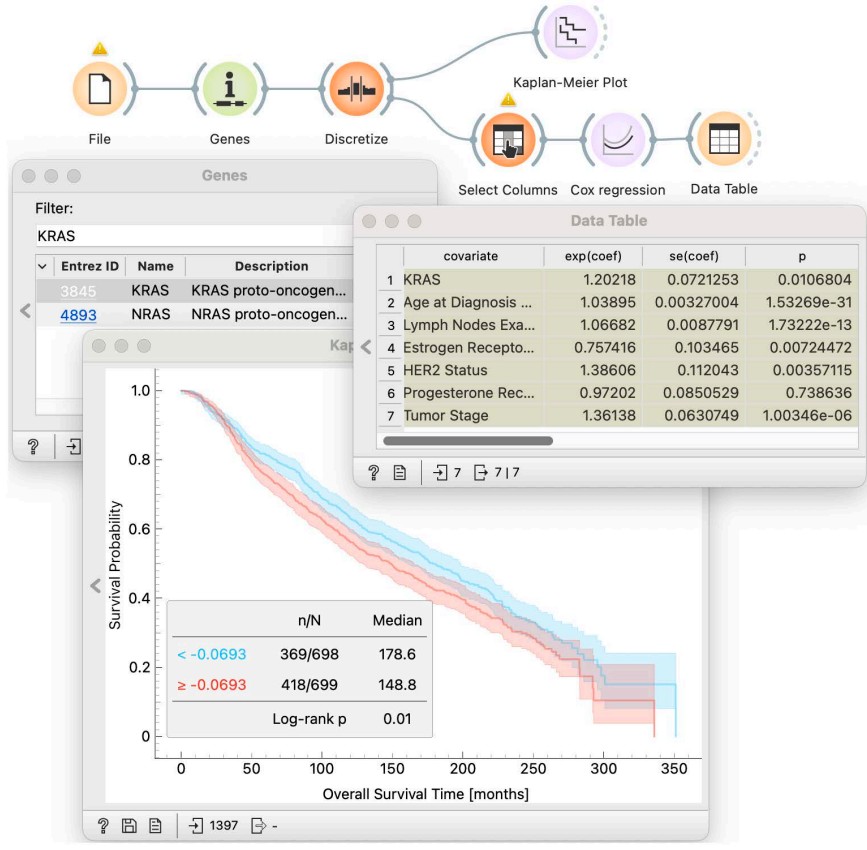

**Fig 6. Evaluation of the prognostic potential of individual genes.** This simple workflow effectively reproduces the study of Hwang et al. [51]. The METABRIC data set is pushed through the *Distributions* widget to allow for an interactive selection of gene expression threshold, and survival probability is visualized in the *Kaplan-Meier Plot* widget. Different gene expression thresholds can be explored at a click of a button.

curves. The lower branch of the workflow performs *Cox regression* analysis for *KRAS* expression levels adjusted for other known clinical factors. Users can compare the results of both analysis side by side. In this workflow we focused only on METABRIC dataset, however, the analysis can be easily repeated with different dataset by loading the data with *File* widget. The analysis above focused only on *KRAS* gene, but how would we identify related genes that may be just as important?

### 5.3 Exploring the related genes

The RAS pathway is a central regulator of cell growth and differentiation in normal tissues, and perturbations of the pathway frequently result in cancer or developmental syndromes (RASopathies) [60]. It is natural to ask the question of the relative importance of other genes in the RAS pathway. Finding such genes can be easily accomplished using a combination of existing and our new widgets (Fig 7).

In Fig 7a, we use *Genes* and *Gene Sets* widgets to pull in the RAS pathway from the KEGG pathway database and keeping only set of genes from the RAS pathway. Like before, we use the *Discretize* widget to set a threshold between high- and low-expressor groups for all the genes in the gene set (**T3**). We use the *Rank Survival Features* widget for ranking genes in the entire gene set and prioritize ones that best separate cohorts according to survival (**T5.1**). We select five top-ranked genes and observe that the gene that is best at separating patients by survival is *FLT3*.

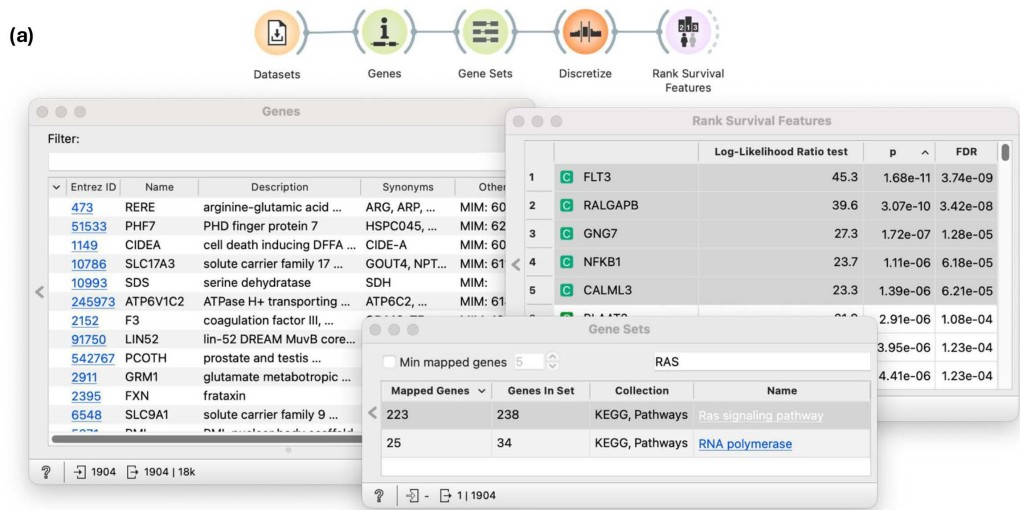

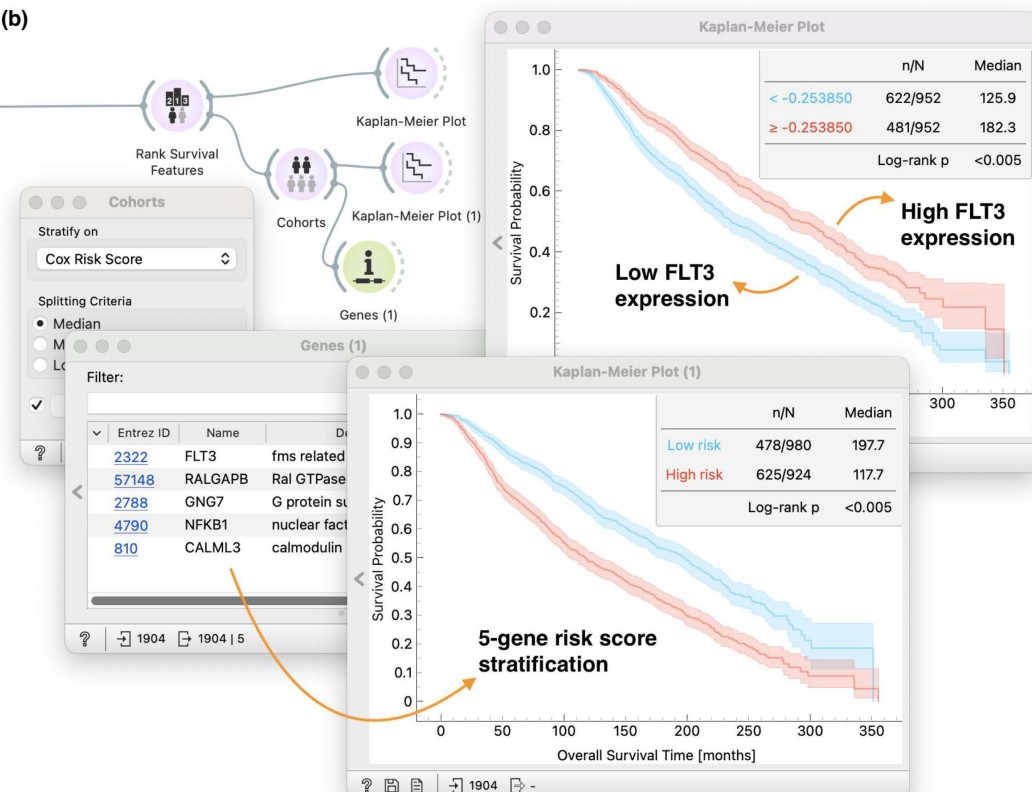

**Fig 7. (a)** By using Gene sets, we have reduced the initial data to include only those genes that are participating in Ras signalling pathway from KEGG pathway databse. **(b)** We conduct standard survival analysis tasks on those genes through *Rank Survival Features* and *Cohorts*. Visualizing survival curves in *Kaplan-Meier plot* allows us to explore newly defined cohorts.

The workflow continues with Fig 7b, we pass the selected genes back into the *Genes* widget to get more context around these genes. For example, we can see that the protein product of *FLT3* expression functions as a receptor tyrosine kinase normally expressed on hematopoietic stem cells. Its mutations are well studied in acute myeloid leukemia [61]. In addition, the upregulation of *FLT3* expression has also been observed in lymph node metastases of patients with primary breast cancer [62]. We observe this also in our exploratory analysis (*Kaplan-Meier plot* in the top branch), where we observe that overexpression of *FLT3* leads to a favorable prognosis (**T1-4**). However, individual genes seldom function in isolation. Understanding the broader molecular circuitry (beyond single-gene biomarkers) can help us better understand the complex nature of diseases such as cancer. The bottom branch of this workflow uses the *Cohorts* widget to develop a risk score model based on all of the selected genes (**T8**). We then use the *Kaplan-Meier Plot* widget to plot and compare survival curves of high- and low-risk groups (**T1-4**). The resulting workflow thus provides a browser of genes and sets of genes that we can evaluate individually or in combination.

### 5.4 Cohort formation based on similarity of data instances

Our exploratory analysis of METABRIC dataset above focused heavily on survival analysis components. We would like to demonstrate that the goals and tasks discussed in Sect 3 can also be used interchangeably with different analysis goals. For example, one might want to explore the similarity of data instances (patients) based on how similar their gene expression profiles are. But then, based on identified clusters, the goal might be to describe and compare different clusters in terms of their survival experiences (**G1 and G2**). The key question is: what characterizes those clusters?

In Fig 8 we demonstrate the utility of *Hierarchical Clustering* to group patients into three distinct clusters. User selected clusters are propagated to *Kaplan-Meier Plot* where we explore the difference of survival functions of patients in different cohorts (**T1-4**). Interestingly, we observe that for the cluster of patients in red survival rates drop rapidly compared to other clusters at the beginning. To investigate further, we extend the workflow and use *Box Plot* to reveal features that characterize patients of each cluster. The exploration of our dataset revealed that patients in the red cluster are defined by the overexpression of gene *PSAT1*. This finding is interesting, since the gene *PSAT1* is indeed a subject of an active research in breast cancer with similar reports [63,64].

Again, notice that any change in interactive cluster selection in *Hierarchical Clustering* propagates through the workflow and updates the content of both *Box Plot* and *Kaplan-Meier Plot* and thus enable on-the-fly hypothesis formation.

## 6 Discussion

Our goal was to design a visual analytics system explicitly tailored to survival data analysis tasks. We aimed to provide comprehensive coverage of key visual analytics concepts (see Sect 2.2) and also to test the hypothesis that a small, well-defined set of reusable visual analytics components could enable domain experts to effectively address a broad range of survival analysis tasks.

Focus of our approach is the flexibility afforded by visual programming, enabling users to construct and adjust data analysis pipelines dynamically. This adaptability supports a nuanced data exploration through comparative analysis and variation of inputs. Changes in the outcomes of the computational analysis may be observed interactively by changing any method's settings, selecting a different data or feature subset, or designing different analysis paths in the same workflow. Our system's design inherently supports the seamless interplay between computational processes and human input and demonstrates how users can be actively engaged in various analytical tasks. In particular, this includes the selection of features and cohorts. Our framework supports various means of human intervention, ranging from computationally driven decisions, such as feature ranking, to domain knowledge-driven decisions, such as selecting genes of interest.

In the manuscript, our proof-of-concept demonstrations focused on showing that the proposed components and user interfaces can readily reproduce analytical steps commonly used in gene expression-based biomarker discovery. However, the proposed tool is primarily designed for new explorations, where fast prototyping and interactive visual analytics

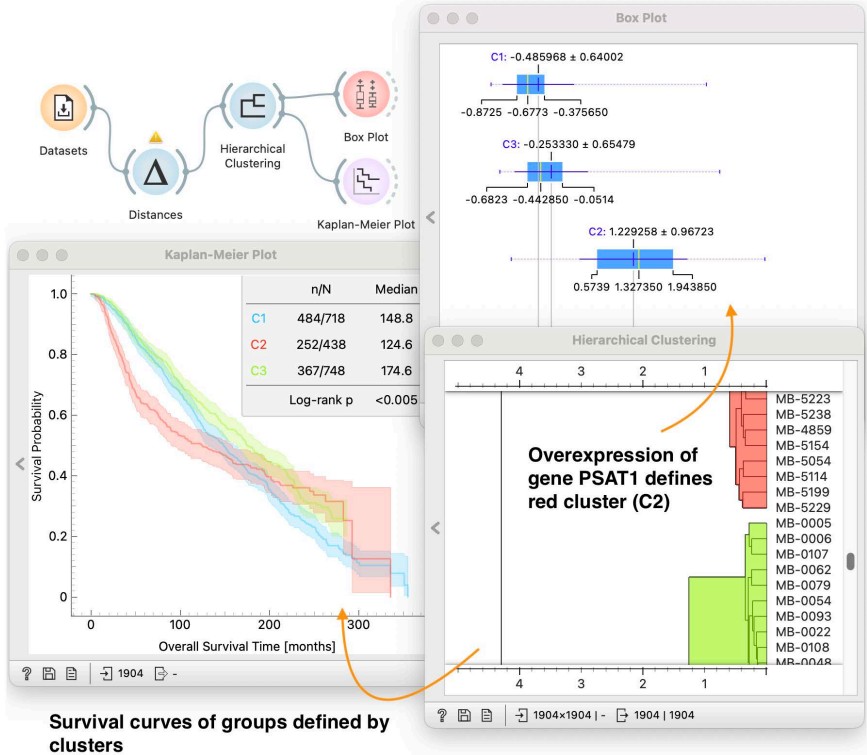

**Fig 8. Identifying groups of patients on the METABRIC dataset.** The workflow computes distances between patient profiles (*Distances* widget) and clusters the patients accordingly in *Hierarchical Clustering*. The user can use the dendrogram in *Hierarchical Clustering* to choose the clusters by brushing or defining a cut-off distance. We display survival curves of identified clusters of patients in the *Kaplan-Meier Plot*. The workflow uses *Box Plot* to characterize given clusters by providing a sorted list of features, which can help in the formation of new hypotheses. In the particular case displayed in the figure, we observe that expression levels of gene *PSAT1* are highest in cluster C2.

become most valuable. To illustrate this, we revisited the SCAN-B breast-cancer dataset from Brueffer et al. [65], which comprises over 3,000 primary tumors profiled by whole-transcriptome sequencing. Whereas the original study focused on training multigene RNA-seq classifiers for conventional clinical biomarkers (for example, predicting estrogen-receptor status from consensus pathology labels), we instead combined their expression and survival data to explore the prognostic potential of small ($N < 10$ genes) interpretable gene sets.

As found in the original analysis of SCAN-B dataset, expression values of single genes that encode the target proteins (e.g., estrogen receptor gene *ESR1*, Fig 9a) cannot stratify patients by survival. We therefore constructed a compact panel of biologically motivated gene sets capturing major breast cancer programs (Estrogen, Proliferation, HER2, PI3K, Immune, EMT), see Fig 9b). Gene-set enrichment scores revealed that Estrogen- and Proliferation-related genes produced the strongest survival separation (Fig 9d, 9e), consistent with extensive literature linking these pathways to patient prognosis [66]. Moreover, projecting gene-set scores into a low-dimensional space (t-SNE) yielded three robust clusters enriched respectively for estrogen signalling (predominantly ER-positive tumors), proliferation/HER2 activity, and EMT/immune features. This construction of the t-SNE plot demonstrates that even a small number of gene sets can expose clinically meaningful structure in the SCAN-B cohort.

This example highlights the strength of our proposed toolbox: users can combine the survival-specific components we developed with the broader machine-learning and bioinformatics methods, like in our case, those on dimensionality reduction, gene-set construction, and ssGSEA scoring (see [67]). Starting from simple domain knowledge-such as an table defining gene

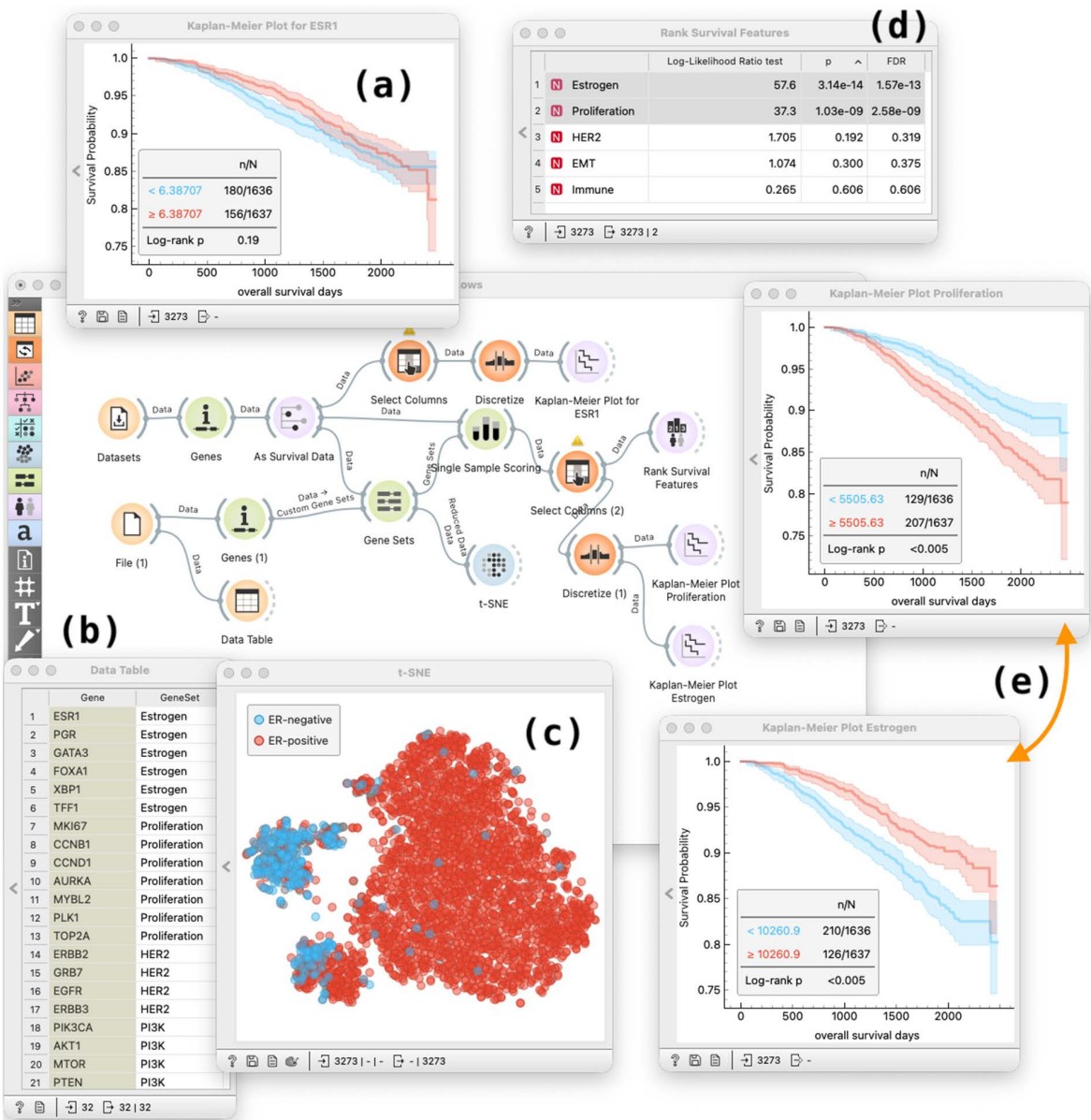

**Fig 9. Workflow for gene set–based biomarker scoring and exploration. (a)** Expression of *ESR1* alone does not sufficiently separate patient survival. **(b)** We load a set of expert-curated gene groups capturing major breast-cancer programs. **(c)** A t-SNE projection of data from gene sets reveals three clusters with different proportions of ER-positive patients. **(d)** Gene sets are ranked by prognostic value, showing Estrogen and Proliferation as most informative. **(e)** A Proliferation and Estrogen-based marker constructed from this sets yields clear separation of Kaplan–Meier survival curves.

sets—a complete exploratory workflow like the one from Fig 9 can be assembled within minutes. The resulting environment encourages rapid hypothesis generation, comparison of analytical choices, and visual discovery of patterns. More broadly, this interoperability demonstrates that our survival analysis framework can be combined with essentially any machine-learning or data-science method, achieving the method integration that was one of the design goals of our approach. This flexibility also means that users can incorporate analytical techniques beyond simple correlation-based exploration. For example, ssGSEA we have included in our use case is a representative of such feature-group modeling approaches.

Support for visual analytics in our proposed framework is primarily a consequence of developing survival analysis components within Orange data mining framework. From Orange, we benefit from interpreted workflows, interactive visualizations, and the ability to encapsulate methods of choice within widgets that act on their own, but through a responsive exchange of data collectively, constrained within a workflow, form an exploratory data analysis environment. This reliance on in-memory processing is what enables such a high degree of interactivity, though it also implies that the framework is best suited for datasets that can fit into the machine's working memory. This design ensures that the analytical environment, characterized by its exploratory nature and the synergy between different components, is entirely shaped by the user's interactions and decisions. While the framework itself does not impose additional scalability constraints, its performance scales with the underlying Python ecosystem that includes NumPy, pandas, and scikit-learn, which allows the toolbox to handle datasets of the same size and complexity supported by these widely used libraries.

In summary, our work contributes to the field of survival analysis by providing a novel visual analytics framework that enhances user engagement through intuitive design and visual programming.

## 7 Conclusion

Survival analysis is concerned with data where the outcome variable of interest is time to an event. Its results can help us understand the effects of factors that determine a patient's lifespan or design treatments for disease. With advances in technology to profile patients, phenotypes, drugs, and treatments and to collect the data related to survival, the democratization of survival data analysis and visual communication of results becomes essential.

In this paper, we have described a tool and an approach for visual analysis of survival data. Our main contributions are the definition and implementation of analytic components for survival analysis workflows and the demonstration that these are sufficient and convenient to solve standard survival analysis tasks. We have also shown that these workflows can bring humans into the loop and thoroughly implement key visual analytics concepts, involving users in both experimentation and interpretation of results.

## Author contributions

**Conceptualization:** Jaka Kokošar, Cagatay Turkay, Luka Ausec, Miha Štajdohar, Blaž Zupan.

**Data curation:** Jaka Kokošar.

**Formal analysis:** Jaka Kokošar.

**Funding acquisition:** Blaž Zupan.

**Methodology:** Cagatay Turkay, Blaž Zupan.

**Resources:** Luka Ausec, Miha Štajdohar.

**Software:** Jaka Kokošar.

**Supervision:** Blaž Zupan.

**Validation:** Jaka Kokošar.

**Visualization:** Jaka Kokošar.

**Writing – original draft:** Jaka Kokošar, Blaž Zupan.

**Writing – review & editing:** Cagatay Turkay, Luka Ausec, Miha Štajdohar, Blaž Zupan.

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
