## [Decision Letter · Decision Letter 0]

17 Sep 2025

Dear Dr. Zupan,

We look forward to receiving your revised manuscript.

Kind regards,

Diego A. Forero, MD; PhD

Academic Editor

PLOS ONE

Journal Requirements:

“This work was supported in part by grants from Slovenian Research Agency (P2-0209, L2-3170, V2-2272).”

“This work was supported in part by grants from Slovenian Research Agency (P2-0209, L2-3170, V2-2272).”

“This work was supported in part by grants from Slovenian Research Agency (P2-0209, L2-3170, V2-2272).”

“This work was supported in part by grants from Slovenian Research Agency (P2-0209, L2-3170, V2-2272).”

5. Thank you for uploading your study's underlying data set. Unfortunately, the repository you have noted in your Data Availability statement does not qualify as an acceptable data repository according to PLOS's standards.

6. We note that Figures 2, 6, 7 and 8 in your submission contain copyrighted images. All PLOS content is published under the Creative Commons Attribution License (CC BY 4.0), which means that the manuscript, images, and Supporting Information files will be freely available online, and any third party is permitted to access, download, copy, distribute, and use these materials in any way, even commercially, with proper attribution. For more information, see our copyright guidelines: http://journals.plos.org/plosone/s/licenses-and-copyright.

a. You may seek permission from the original copyright holder of Figures 2, 6, 7 and 8 to publish the content specifically under the CC BY 4.0 license.

Reviewers' comments:

Reviewer's Responses to Questions

**Comments to the Author**

1. Is the manuscript technically sound, and do the data support the conclusions?

Reviewer #1: Yes

Reviewer #2: Yes

2. Has the statistical analysis been performed appropriately and rigorously?

Reviewer #1: Yes

Reviewer #2: Yes

3. Have the authors made all data underlying the findings in their manuscript fully available?

Reviewer #1: Yes

Reviewer #2: No

4. Is the manuscript presented in an intelligible fashion and written in standard English?

Reviewer #1: Yes

Reviewer #2: Yes

Reviewer #1: This article presents a visual analysis methodology designed to simplify survival analysis in biomedicine. It proposes a framework of reusable visualization and modeling components, akin to "LEGO bricks," to aid in biomarker discovery. This approach addresses the limitations of existing methods that often require programming skills or are inflexible, thus hindering accessibility for biomedical researchers. Its goal is to provide a user-friendly, open-source tool (implemented as an Orange Data Mining add-on) that facilitates interactive data exploration and hypothesis-driven research. It is relevant for the journal. However, I have some suggestion for improvement shown below.

1. The article provides a valuable tool for survival data analysis, but could be enhanced by explicitly addressing scalability for large and complex datasets.

2. Future improvements could focus on advanced interactive visualization techniques to uncover intricate relationships within high-dimensional data, potentially incorporating machine learning-driven insights for predictive modeling and automated pattern detection.

3. Expanding the tool's interoperability with other popular biomedical data platforms would broaden its adoption.

4. A detailed discussion of how the tool supports causal inference in survival analysis, beyond mere correlation, would significantly enhance its analytical capabilities.

5. Please improve the quality of the figures.

Reviewer #2: The paper describes a visual analytics tool for survival analysis. It presents substantial work and is potentially interesting for a broad readership. The paper is well-written and easy to follow. My major concern is the novelty of the work. As contribution, the authors mention support for cohort definition and cohort discovery. Highly relevant papers in this area (also with medical applications), however, are not even cited:

Supporting iterative cohort construction with visual temporal queries

J Krause, A Perer, H Stavropoulos

IEEE transactions on visualization and computer graphics 22 (1), 91-100

Interactive visual patient cohort analysis

Z Zhang, D Gotz, A Perer

Proc. of IEEE VisWeek Workshop on Visual Analytics in Health Care

Somarakis, A., Ijsselsteijn, M. E., Luk, S. J., Kenkhuis, B., de Miranda, N. F., Lelieveldt, B. P., & Höllt, T. (2020). Visual cohort comparison for spatial single-cell omics-data. IEEE Transactions on Visualization and Computer Graphics, 27(2), 733-743.

Preim, B., Klemm, P., Hauser, H., Hegenscheid, K., Oeltze, S., Toennies, K., & Völzke, H. (2016). Visual analytics of image-centric cohort studies in epidemiology. In Visualization in Medicine and Life Sciences III: Towards Making an Impact (pp. 221-248). Cham: Springer International Publishing.

Thus, the contributions should be either better justified or revised.

Author summary:

"many clinicians and life science researchers" - I would remove "clinicians". Without very few exceptions,

clinicians, i.e., the persons treating patients under severe time pressure, is not the target group for this research.

I also wonder whether the title of the paper is ideal. Of course, the method is not restricted to cancer, not even to medicine, but cancer patients is by far the most important application, also discussed in the paper. Reading the abstract, I thought, what this might be good for and clearly for cancer patients. But cancer is neither part of the abstract, nor the title. The authors should at least consider to change this.

I would be more convinced of the paper if the tool indeed helped to gain new insights. I like the idea of insight-based evaluations (Chris North) for visual analytics applications. I would not say that such an evaluation is mandatory for publishing this work, but to better understand examples of true insights would be helpful. I understand that the author did a lot to apply their tool to real world data, also to confirm existing hypotheses - that is also useful, but I miss the aha-moment: Here is something that is very hardly possible with any other tool; the uniqueness. Applying visual analytics to e.g., gene-related data - that is really not new.

The terms "clinical event of interest" and "clinical outcomes" have strongly overlapping, if not identical meanings. I recommend to either clarify what discriminates both terms or use just one of them.

The term "interpreted workflow" is interesting. I suggest to emphasize it and elaborate it.

Figure 3: I would use the term brush-and-link at least in "()" in the caption.

References: A couple of conference references may be shortened, e.g., 18, 21, 23. They contain the year twice and IEEE twice.

**Do you want your identity to be public for this peer review?** For information about this choice, including consent withdrawal, please see our Privacy Policy

Reviewer #1: **Yes:** Sudhir K. Routray

Reviewer #2: No

---

## [Author Response · Author response to Decision Letter 1]

25 Nov 2025

We would like to thank the editor and the reviewers for the feedback on our previous submission, which helped us to improve the manuscript. We provide point-by-point responses to the reviewers' suggestions for changes in an accompanying document.

---

## [Decision Letter · Decision Letter 1]

6 Jan 2026

Visual analytics framework for survival analysis and biomarker discovery from gene expression data

PONE-D-25-25571R1

Dear Dr. Zupan,

We’re pleased to inform you that your manuscript has been judged scientifically suitable for publication and will be formally accepted for publication once it meets all outstanding technical requirements.

Kind regards,

Diego A. Forero, MD; PhD

Academic Editor

PLOS One

Additional Editor Comments (optional):

Reviewers' comments:

Reviewer's Responses to Questions

**Comments to the Author**

Reviewer #2: All comments have been addressed

Reviewer #3: All comments have been addressed

2. Is the manuscript technically sound, and do the data support the conclusions?

Reviewer #2: Yes

Reviewer #3: Yes

3. Has the statistical analysis been performed appropriately and rigorously?

Reviewer #2: Yes

Reviewer #3: Yes

4. Have the authors made all data underlying the findings in their manuscript fully available?

Reviewer #2: Yes

Reviewer #3: Yes

5. Is the manuscript presented in an intelligible fashion and written in standard English?

Reviewer #2: Yes

Reviewer #3: Yes

Reviewer #2: The paper is now in a very good shape. I only found a few minor spelling issues:

Line 354: a interactive -> an

Line 441: user selected -> user-selected

Line 486: Is the term "breast cancer programme" appropriate here? I know this term in a different meaning.

Line 499: an table defing -> a table

Reviewer #3: This is a strong and timely contribution that introduces a flexible, interactive, and accessible visual analytics framework for survival analysis and biomarker discovery. The manuscript is clearly written, methodologically rigorous, and supported by well-chosen case studies.

Major strengths

Methodological innovation:

The task-oriented abstraction of survival analysis goals and their implementation as reusable visual components is elegant and convincing.

Usability and accessibility:

The visual programming paradigm lowers the barrier for clinical and biomedical researchers, addressing a genuine gap in current survival analysis practice.

Reproducibility and transparency:

Open-source availability of the tool and workflows aligns well with open science principles and enhances the impact of the work.

Well-chosen case studies:

Use of METABRIC and SCAN-B datasets effectively demonstrates both replication of published analyses and exploratory discovery.

Suggestions for improvement (minor revisions)

Exploratory vs confirmatory analysis

Given the interactive and exploratory nature of the framework, it would be helpful to more explicitly discuss:

risks of multiple testing,

data-driven cut-off selection,

and potential inflation of false-positive findings in biomarker discovery.

A short clarification in the Discussion would be sufficient.

Generalizability beyond genomics

While the cancer gene-expression examples are compelling, the authors may briefly comment on applicability to: non-omics clinical covariates, treatment groups, or composite clinical risk scores.

Scalability considerations

The manuscript mentions in-memory processing constraints; adding approximate practical limits (e.g., number of features or samples) would help readers assess feasibility for large-scale datasets.

Comparative summary

A concise table comparing the proposed framework with existing web-based survival tools (e.g., in terms of interactivity, transparency, workflow flexibility) could further clarify the unique contribution.

Overall, this is a high-quality manuscript, and the suggested points are intended to strengthen clarity rather than address fundamental weaknesses. I commend the authors for a thoughtful and well-executed study.

**Do you want your identity to be public for this peer review?** For information about this choice, including consent withdrawal, please see our Privacy Policy

Reviewer #2: No

Reviewer #3: No

---

## [Editor Report · Acceptance letter]

PONE-D-25-25571R1

PLOS One

Dear Dr. Zupan,

I'm pleased to inform you that your manuscript has been deemed suitable for publication in PLOS One. Congratulations! Your manuscript is now being handed over to our production team.

Kind regards,

on behalf of

Dr. Diego A. Forero

Academic Editor

PLOS One